# Electrically-Induced Ventricular Fibrillation Alters Cardiovascular Function and Expression of Apoptotic and Autophagic Proteins in Rat Hearts

**DOI:** 10.3390/ijms20071628

**Published:** 2019-04-02

**Authors:** Andras Czegledi, Agnes Tosaki, Alexandra Gyongyosi, Rita Zilinyi, Arpad Tosaki, Istvan Lekli

**Affiliations:** 1Department of Pharmacology, Faculty of Pharmacy, University of Debrecen, 4032 Debrecen, Hungary; czegledi.andras@pharm.unideb.hu (A.C.); gyongyosi.alexandra@pharm.unideb.hu (A.G.); zilinyi.rita@pharm.unideb.hu (R.Z.); lekli.istvan@pharm.unideb.hu (I.L.); 2Dermatology Clinic, University of Debrecen-Medical Center, 4032 Debrecen, Hungary; tosakiagnes@med.unideb.hu

**Keywords:** ventricular fibrillation, apoptosis and autophagy, cell deaths

## Abstract

Background: The pathological heart contractions, called arrhythmias, especially ventricular fibrillation (VF), are a prominent feature of many cardiovascular diseases leading to sudden cardiac death. The present investigation evaluates the effect of electrically stimulated VF on cardiac functions related to autophagy and apoptotic mechanisms in isolated working rat hearts. Methods: Each group of hearts was subjected to 0 (Control), 1, 3, or 10 min of spacing-induced VF, followed by 120 min of recovery period and evaluated for cardiac functions, including aortic flow (AF), coronary flow (CF), cardiac output (CO), stroke volume (SV), and heart rate (HR). Hearts were also evaluated for VF effects on infarcted zone magnitude and Western blot analysis was conducted on heart tissue for expression of the apoptotic biomarker cleaved-caspase-3 and the autophagy proteins: p62, P-mTOR/mTOR, LC3BII/LC3BI ratio, and Atg5-12 complexes. Results: Data revealed that VF induced degradation in AF, CF, CO, and SV, which prominently included-variable post-VF capacity for recovery of normal heart rhythm; increased extent of infarcted heart tissue; altered expression of cleaved-caspase-3 suggesting potential for VF-mediated amplification of apoptosis. VF influence on expression of p62, LC3BII/LC3BI, and Atg5-12 proteins was complex, possibly due to differential effects of VF-induced expression on proteins comprising the autophagic program. Conclusions: VF was observed to cause time-dependent changes in autophagy processes, which with additional analysis under ongoing investigations, likely to yield novel therapeutic targets for the prevention of VF and sudden cardiac death.

## 1. Introduction

Several pathological processes to cardiovascular tissue occur as a result of sudden restoration of blood flow and cardiac function to tissues, and such an event may result in characteristic deleterious effects and symptoms that include impairment of physiological functions. These outcomes are due to oxidation of myocardial membrane lipids, perturbed ion homeostasis, and reactive oxygen species [1,2,3,4], which lead to various cell deaths, including necrosis, apoptosis, and autophagy. Support for this mechanistic model is provided by an observation that tissue damage and resulting arrhythmias may be limited by pharmacological interventions diminishing their contribution to the arrhythmic component of cardiac failure [5,6,7,8,9]. Previously, we have provided evidence of the above processes as major underlying contributors to the pathophysiology of cardiac arrhythmias, particularly ventricular fibrillation (VF) [10,11]. Arrhythmias are a major cause of mortality in patients afflicted with cardiovascular disease, or who have sustained cardiac injury sufficient to trigger these phenomena [12,13], particularly those, in which persistent angina pectoris is a major symptom [14] and may also be triggered by thrombolytic therapy, saphenous vein bypass grafting, release of coronary spasm, and angioplasty [15]. Specific cellular events resulting in arrhythmias remain an open area for investigation. Evidence indicates that cell membrane damage in cardiomyocytes results in the loss of stable compartmentalization and transmembrane flux Ca^2+^, K^+^, Mg^2+^, and Na^+^ [16] with consequent derangement of coordinated signaling to contractile elements of the myocardium, which leads to ventricular fibrillation [11]. A major question that remains substantially unanswered is the extent to which arrhythmia-associated pathophysiological changes to cardiovascular cells affect adaptive processes that influence likelihood of recovery, versus death or debilitation following an arrhythmic event. The present investigation addresses this issue by examination of electrically-induced VF on cardiac function, magnitude of infarct zone extent, and expression of proteins that are clinical and/or objective laboratory biomarkers for normal, versus disrupted heart function, necrosis, apoptosis, and autophagy. It has previously been demonstrated that alterations in the principle cardiac functions, including aortic flow (AF), coronary flow (CF), cardiac output (CO), stroke volume (SV), and heart rate (HR) correlate with occurrence of arrhythmias in ways that clearly demonstrate common underlying regulatory processes for both cardiac functions and arrhythmogenesis [17,18,19]. Nevertheless, it is presently unknown whether various pathologic processes, which result in disruption of physiological heart function and contributing cellular processes, such as apoptosis and autophagy, may be promoted by the occurrence of arrhythmias. The major hypothesis of the present investigation is that the occurrence of VF, which is one of the most clinically significant arrhythmic disorders leading to sudden cardiac death, directly influences apoptotic and autophagic processes and signal transductions to an extent that arrhythmias may act as major inducers of alterations in basic cellular, tissue, and organ processes that underlie myocardial diseases.

## 2. Results

### 2.1. Effects of Pacing Induced Ventricular Fibrillation on Cardiac Recovery

The time-dependent effects of electrically-induced VF, perfusion and post-fibrillation recovery intervals, on cardiac function in isolated working hearts are shown in Figure 1. As shown in Figure 1A. AF was significantly reduced relative to the time-matched control values for hearts subjected to 10 min of electrically-induced VF followed by 30, 60, and 120 min of post-VF recovery (*p* = 0.0175). Interestingly, the average CF magnitudes (Figure 1B), in hearts subjected to 10 min of electrical fibrillation induction, were significantly elevated relative to the control values following 60 min and 120 min of recovery, respectively (*p* = 0.0149). Outcomes of cardiac output (CO) assessment shown in Figure 1C. indicates a significantly reduced CO magnitude after a 1, 3, and 10 min period of VF followed by 30, 60, and 120 min of aerobic perfusion at each time point, respectively, (*p* = 0.0432). A reduction in SV values after 10 min of VF followed by 120 min of aerobic perfusion (Figure 1D) was observed. No substantial VF stimulation time-dependent effects were observed for either stroke volume Figure 1E. and heart rate Figure 1F. relative to the time-matched control values at the corresponding time points.

### 2.2. Induced Ventricular Fibrillation Effects on the Infarcted Zone Magnitude

As it can be seen in Figure 2, no significant alteration in infarct volume was observed in hearts undertaken for 1 and 3 min of pacing induced VF followed by 120 min of aerobic perfusion, in comparison with hearts that received no electrically induced VF (control). Nevertheless, the average infarcted size magnitude in hearts subjected to 10 min of VF stimulation was approximately 10-fold higher compared to the control (no electrically induced VF) group (*** *p* < 0.001).

### 2.3. Induced Fibrillation on Caspase-3 Expression and TUNEL Positivity

Figure 3A. shows that relative to hearts subjected to no VF (control), and hearts subjected to 1 min of VF expressed approximately 1.5× the quantity of cleaved-caspase-3 (* *p* < 0.05); and approximately double the quantity of this protein after 3 min of VF (** *p* < 0.01); but with no significant difference from the control hearts after 10 min of VF period followed by 120 min of aerobic perfusion. In line with the caspase-3 expression, hearts subjected to 1 min and 3 min of VF followed by 120 min of perfusion exhibited significant TUNEL positivity (Figure 3B). The most intense TUNEL positivity was observed in hearts undertaken to 3 min of VF followed by 120 min of post-fibrillated perfusion (Figure 3B). The numeric values of TUNEL positive nuclei are depicted in Figure 3C, showing that TUNEL positivity was significantly increased in all three groups (*** *p* < 0.001) in comparison with the control value.

Figure 4A shows p62 in hearts subjected to 1, 3, and 10 min VF stimulation periods followed by 120 min perfusion and expressed by approximately 1/3 (** *p* < 0.01), 1/2 (*** *p* < 0.001), and 2/3 (** *p* < 0.001) less of this protein, respectively. Expression levels of P-mTOR/mTOR ratio were determined as a function of VF stimulation periods and displayed in Figure 4B. The ratio of P-mTOR/mTOR showed a gradual increase with the extended duration of electrical VF. It is also shown in Figure 4B. that relative to the unstimulated control group, no significant difference for the average ratio of P-mTOR/mTOR ratio was observed in hearts subjected to 1 min of electrical VF. This value was approximately 1/4 higher in hearts subjected to 3 min of VF period (* *p* < 0.05) and further increased in hearts subjected to 10 min of VF (* *p* < 0.05). The time-dependent effect of VF on the expression of LC3BII/LC3BI protein ratio by tissue from isolated perfused hearts is shown in Figure 4C. The results revealed that relative to the unstimulated Control group, hearts subjected to 1, 3, and 10 min of VF periods showed no significant changes in this protein expression, although a gradual decrease was registered. The average expression levels of Atg5-12 complex proteins as a function of VF stimulation is shown in Figure 4D. Relative to the control group receiving no VF stimulation, no significant differences in average expression of this protein complex were observed for hearts subjected to 1 min and 10 min periods of VF. However, interestingly, the average Atg5-12 expression levels in hearts subjected to 3 min of stimulated VF were approximately increased by 25% (*p* < 0.05) in comparison with the non-fibrillated control group.

## 3. Discussion

### 3.1. Pacing-Induced VF on Infarct Size and Caspase-3 Activation

The effect of ventricular arrhythmias on cardiac function and different pathways of cell death were the main focus in the present investigation. The duration of electrical stimuli induced ventricular fibrillation could be contributed to the depression of cardiac function including aortic flow, coronary flow, and stroke volume changes. Decreased cardiac function was accompanied by enlarged infarcted volume. Compared to the Control hearts subjected to no VF stimuli, the approximately 10-fold greater infarcted zone extent was detected in hearts subjected to 10 min duration of VF, which may be accounted for the capacity of electrically-induced VF to promote pathway-mediated oxidative damage in heart tissue as described earlier [18]. The longest duration of electrically-induced VF was detrimental to the development of infarcted myocardium as TTC staining discriminates between the living and dead tissues [20,21]. The known mechanisms by which infarcted zone expansion is triggered and augmented, strongly suggests degradation of metabolite compartmentalization within cardiomyocytes and other cardiovascular cells, resulting in both arrhythmogenesis and derangement of cardiac function [22,23]. An enhanced number of TUNEL positive cells were observed in all electrically fibrillated hearts. Interestingly, it was the highest following 3 min period of electrical fibrillation. These observations were in line with the cleaved caspase-3 results indicating the highest cleaved caspase-3 expression in hearts undertaken to 3 min of ventricular fibrillation. Caspase-3 activation is an early event in the apoptotic cell death cascade, and very high activity of this enzyme may trigger disease through excessive apoptosis in the intact myocardium [24]. Thus, the presence of the cleaved molecule could be a major biomarker for cardiovascular pathologies, prominently in stroke and myocardial infarction, leukemia [25], and death of neurons in Alzheimer’s disease [26]. Authors have previously investigated various processes and demonstrated how amplification of endogenous cytoprotection by induction of heme oxygenases-1 (HO-1) using sour cherry seed-derived HO-1 inducers, and/or beta carotene dramatically limits the extent of infarcted zones in cardiac tissue [27,28].

### 3.2. VF Influence on Apoptosis- and Autophagy-Associated Proteins: Western Blot Studies

Western blot data in Figure 3. shows VF effects on expression of apoptosis, and it reveals that the duration of VF significantly influenced the level of cleaved caspase, at which cardiomyocytes experienced during these processes in the model used under in vitro conditions. Figure 3A. shows significant VF-dependent increases in tissue production of cleaved (activated) caspase-3 (cysteinyl aspartate-specific protease-3) at 1 and 3 min of VF, relative to the control value. The ubiquitin-binding p62 sequestosome 1 protein (SQSTM1) accumulates when autophagy is inhibited, thus, the significantly decreased p62 expression in isolated working heart tissue with 1–10 min of VF relative to the control value, indicates that autophagy is induced by VF, as shown in Figure 4A. Outcomes of analysis for phosphorylated mammalian target of rapamycin (P-mTOR), which is the activated form of this enzyme on this cell growth in response to nutritional cues [29], and its ratio to the inactive form (P-mTOR/mTOR) is shown in Figure 4B. Here, the significantly increased P-mTOR/mTOR ratio occurring with longer VF periods is consistent with the expected result of exposure of cells to elevated autophagic processes. It is known that oxidative damage to a cell’s genome induces increased high levels of mTOR activation, and cell cycle arrest (an anticancer mechanism), which subsequently also causes formation of senescent cell phenotypes that are hypersecretory for inflammatory cytokines, an effect that causes inflammatory tissue damage—constitution a major underlying mechanism of age-related physical deterioration [30]. Authors of this report have previously demonstrated that the pathophysiology of these processes is promoted by cardiac arrhythmias and counteracted by induction of autophagy [31,32]. These findings further reinforce observations shown in Figure 4A. demonstrating VF-associated decrease in cardiac tissue expression of the p62 protein. p62 acts as a sensor molecule recognizing toxic cellular waste and is subsequently sequestered into autophagosomes resulting in its rapid clearance in the cell, thus decreasing its signal activity [33].

### 3.3. Usefulness and Limitations on Use of LC3BII/LC3BI and Atg5-12 Complex Proteins as Autophagy Indicators in Fibrillated Hearts

Outcomes of the final set of experiments conducted in this investigation are shown in Figure 4C,D. Results in Figure 4C show a gradually decreased but not significant in LC3BII/LC3BI expression in all groups compared to the control value, which suggests that VF induction suppresses the autophagy processes. Thus, this report provides preliminary evidence for the validity of this hypothesis [32]. The possibility that hearts, at least within the constraints of the present model, may exhibit characteristic fluctuations in VF-associated autophagic responses, is further reinforced by the results of assay for Atg5-12 complex proteins shown in Figure 4D. Initially, with the context of results shown in Figure 4C, the time course of LC3BII/LC3BI expression suggests a VF-induced decrease in autophagy, followed by restoration to the levels exhibited by the control group. In addition, data in Figure 4A, shows a progressive decrease in the p62 expression signal during 1–10 min of VF, suggesting that autophagy is promoted. These outcomes notwithstanding, the results shown in Figure 4D should be interpreted with an understanding of the biological role of the Atg5-12 complex proteins. These proteins interact in fairly complex ways in autophagosome assembly, with expression, function, and degradation of each component occurring according to a tightly coordinated program over the timeline, in which a cell or tissue is undergoing an autophagic response [34]. Thus, it is likely that the highly prooxidant tissue environment of the myocardium experiencing electrically-stimulated VF would experience significant derangement in the expression and degradation kinetics of individual Atg proteins, rendering the Atg5-12 signal unsuitable as a single and simple metric for autophagy. Data in Figure 4C,D may support this possibility by the existence of fluctuation in expression of each of the proteins studied. Here, their presence in the tissue is clearly related to the effect and role of VF on autophagy, but might be more relevant to VF-induced influence on particular protein components than on the predominant occurrence of autophagy in VF-treated heart tissue. However, it has to be noted that in the present study autophagic flux was not studied.

### 3.4. Conclusions

This investigation has revealed several features of heart tissue response to electrically-stimulated VF that may prove enormously useful in ongoing exploration of arrythmogenesis, with particularly valuable insight into autophagy-related pathophysiology. A major positive outcome, as this has improved understanding of both the usefulness and limitations of the electrically-stimulated VF model of the isolated working rodent heart, in which the authors have relied on extensively as an investigative tool over several decades of cardiovascular research. It is here acknowledged that although some of its aspects may depart from being precisely analogous to in vivo conditions, the tissue environment and resulting arrhythmogenic consequences provided a close enough parallel to the failing vertebrate heart as to allow clinically relevant conclusions to be drawn. Further, results of this report which demonstrated variable post-VF capacity for recovery of normal rhythm from heart to heart, underscored the caution that cardiologists may need to exercise in tailoring therapy to individual patients. Observations of VF effects on cardiac infarction expanded the scope of potential use for inducers of endogenous defenses including apoptosis and autophagy as a countermeasure to this effect; and demonstration of VF effects on biomarkers for apoptosis, and autophagy-associated proteins open novel directions in approaches to intervention in arrhythmogenic influences by which each process may be manipulated to address a wide range of disease states, extending on cardiovascular medicine.

## 4. Materials and Methods

### 4.1. Animals

Male Sprague-Dawley rats weighing 370–400 g (Charles River International, Inc., Sulzfeld, Germany) were used for all experiments in the present study. The animals were housed at room temperature (approximately 25 °C), with lighting set to alternating dark and light periods of 12 h. The rats were maintained on normal rodent chow and tap water ad libitum. All animals received humane care, in compliance with the “Principles of Laboratory Animal Care” according to the National Society for Medical Research and the Guide for the Care and Use of Laboratory Animals, formulated by the National Academy of Sciences, and published by the National Institute of Health (NIH Publication No. 86-23, revised 1985). All protocols involving animal use were approved by the Committee of Animal Research, University of Debrecen, Hungary (3/2012/DEMAB, 21-03-2012).

### 4.2. Chemical Reagents

All chemicals and buffer solutions were obtained from Sigma-Aldrich Ltd. (Schnelldorf, Germany). The perfusion medium was a modified Krebs-Henseleit bicarbonate buffer composed of NaCl 118 mmol, KCl 5.8 mmol, CaCl_2_ 1.8 mmol, NaHCO_3_ 25 mmol, NaH_2_PO_4_ 0.36 mmol, MgSO_4_ 1.2 mmol, and glucose 11.1 mmol, dissolved in redistilled water.

### 4.3. Isolated Working Heart Preparation and Perfusion Protocol

Rats were anesthetized with intraperitoneal injection of ketamine (50 mg/kg) and xylazine (10 mg/kg), and heparin (1000 U/kg) was then injected intravenously through the dorsal penile vein. After thoracotomy, hearts were excised, and placed into ice-cold perfusion buffer, with a composition described above (chemical reagents) and filtered through a 0.65 μmol porosity filter to remove any particulate contaminants. Perfusion of hearts was conducted using a 5 min washout period in Langendorff mode via the cannulated aorta, and then the left atrium was also cannulated. The perfusion fluid then passed to the left ventricle, from which it was spontaneously ejected through the aortic cannula against a pressure equivalent to 100 cm of water [18].

### 4.4. Test Groups

Hearts taken from the animals were segregated into test groups and subjected to 0 (Control), 1, 3, and 10 min of VF periods, respectively. Electrical fibrillation was carried out by 20 Hz (1200 beats/min) for 1, 3, and 10 min, respectively, using 5 V square-wave pulses of 2 ms duration. The active lead was attached to the apex of the heart and the ground lead was attached to the aortic cannula. These conditions suppressed the sinus node activity and resulted in VF consisting of an irregular undulating baseline on the ECG. Following the termination of electrical fibrillation, hearts were allowed post-fibrillation recovery period for 120 min. At the end of the 120 min of post-fibrillation recovery period, the expression or repression of the autophagic and apoptotic proteins were determined. Aortic flow, coronary flow, cardiac output, stroke volume, and heart rate were monitored. The aortic flow and coronary flow rate were measured by timed collection of the coronary perfusate that dripped directly from each heart. The cardiac output was determined as the sum of aortic flow and coronary flow in milliliters per min, from which stroke volume was additionally calculated.

### 4.5. Infarction Zone Magnitude in VF-Treated Isolated Hearts

Triphenyl-tetrazolium chloride (TTC) staining was used to determine infarcted areas. For these experiments, hearts were subjected to 0 (Control), 1, 3, and 10 min of VF followed by 120 min of post-fibrillated perfusion. A total of 100 mL of 1% TTC solution in phosphate buffer (Na_2_HPO_4_, 88.05 mmol, NaH_2_PO_4_ 1.8 mmol) was heated to 37 °C and administered directly via the side arm of the aortic cannula. TTC-stained viable myocardium was deep red, while potentially infarcted areas remained pale, allowing clear demonstration and measurement of infarction magnitude. Hearts were stored at −80 °C for a subsequent analysis and then sliced transversely to the apico-basal axis into 2–3 mm thick sections, weighted, blotted dry, and placed in between microscope slides, then scanned at 600 dpi on a Hewlett-Packard Scanjet single pass flatbed scanner (Hewlett-Packard, Palo Alto, CA, USA). Using ImageJ software, each image was subjected to equivalent degrees of background subtraction, brightness, and contrast enhancement for improved clarity and distinctness. Infarct area (pale areas, white pixels) of each slice were traced and the respective areas were calculated by pixel density analysis. Infarcted areas and total area were measured by computerized planimetry software. Infarct size was expressed as a percentage ratio of the infarcted zone to the total area in each heart (percentage of pixels).

### 4.6. Westerns Blot Analysis

Protein expression levels for cleaved-caspase-3, p62, mTOR, phosphorylated P-mTOR, LC3BII/LC3BI, and Atg5-12 complex proteins were evaluated using Western immunoblotting. Proteins were isolated from myocardial tissue with homogenizing buffer (25 mM Tris-HCl, pH 8, 25 mmol NaCl, 1 mmol Na-Orthovanadate, 10 mmol NaF, 10 mmol Na-Pyrophosphate, 10 nmol Okadiac acid, 0.5 mmol EDTA, 1 mmol PMSF, 1× protease inhibitor cocktail) using a polytron homogenisator on ice. Next, after 10 min centrifugation, 2000 rpm at 4 °C supernatants were transferred to fresh tubes and centrifuged for 20 min at 10000 rpm and 4 °C. The supernatants were used as cytosolic fractions. Protein concentration was measured by BCA Protein Assay Kit (Thermo Scientific, Waltham, MA, USA). Proteins were separated using 4–20% Mini-PROTEAN® TGX Stain-Free™ Protein Gels then transferred onto polyvinylidene fluoride (PVDF) membranes. Non-specific binding was blocked using non-fat milk in Tris-buffered saline with 1% Tween20 for 30 min at room temperature. After washing with TBS-T the membranes were incubated overnight at 4 °C with primary antibodies against LC3B (1:1000), Atg5 (1:500), Atg12 (1:500), p62 (1:1000), cleaved-caspase3 (1:1000), mTOR (1:2000) and P-mTOR (1:2000). After 3 washes the membranes were incubated with secondary antibody for 1.5 h at room temperature and signal intensities for each protein band were detected using Clarity Western ECL Substrate (Bio-Rad, Hercules, CA, USA). The optical density of bands was measured using the ChemiDoc Touch Imaging System and protein levels determined with the Bio-Rad Image Lab 5.2 software (Bio-Rad). The levels of the investigated proteins were normalized against the total protein loaded on the gels. The protein expression was quantified by the ratio of (band volume)/(total protein volume). Thus, this method eliminates the need for housekeeping proteins [35,36].

### 4.7. Fluorescens Cell Death Detection

To detect apoptosis, we used the terminal deoxynucleotidyl transferase (TdT) nick end labelling test by the In Situ Cell Death Detection Kit, TMR (fluorescein-labeled cell markers) red (Roche, Mannheim, Germany). Apoptosis can be detected by labeling the free 3′-OH termini with modified nucleotides in an enzymatic reaction. The enzyme terminal deoxynucleotidyl transferase (TdT) catalyzes the template-independent polymerization of deoxyribonucleotides to the 3′-end of single- and double-stranded DNA. Harvested hearts tissue were fixed in 4% formalin for 24 h at 4 °C, embedded in paraffin, and cut into 4.5 µm thick sections. All tissue sections were deparaffined in xylene and acetone then rehydrated in 70% ethanol and water. The sections were boiled in citrate buffer pH 6.0 for 12 min, then cooled at room temperature for 20 min, thereafter, washed two times for 5 min in phosphate-buffered saline (PBS pH 7.4). Finally, sections were incubated with TdT (terminal deoxynucleotidyl transferase) in a humidified box, at 37 °C for 1 h. After washing, to identify nuclei, we used DAPI (4′,6-diamidino-2-phenylindole), which emits blue fluorescence upon binding to AT regions of DNA (Thermo Fisher Scientific, Waltham, MA, USA). The slides were then washed with PBS, after being air-dried, and subsequently covered with mounting medium and glass slide covers. Moviol solution was used as a mounting medium. Fluorescence microscopic images were obtained by a Zeiss AxioScope A1 microscope, EC Plan-Neo fluar 40×/0.75 M27 objective lens with HBO100 illuminator and Zeiss AxioCam ICm1 camera (Zeiss, Jena, Germany). After merging the blue and red channels, purple spots were associated with apoptotic nuclei, while blue spots were identified as non-apoptotic nuclei (ZEN 2012 software, Zeiss). Apoptosis was quantified by ratio of TdT-positive nuclei/total nuclei in each section.

### 4.8. Data Analysis and Statistics

All data are presented as average magnitudes of each outcome in a group ± standard error of the mean (SEM). Normality of data sets were verified with the D’Agostino-Pearson omnibus test. A Student’s *t*-test was used to determine differences between two sets of data. More than two data sets were compared with one-way ANOVA followed by Tukey post-testing. Difference of means was considered significant at * *p* < 0.05, ** *p* < 0.01, and *** *p* < 0.001. Statistical analysis was carried out with GraphPad Prism 6.07. (GraphPad Software Inc., San Diego, CA, USA).

## 5. Limitation of the Study

It has been noted that direct measurement of the oxidative stress marker was not assessed. However, previously it has been shown that pacing-induced VF induces free radical formation [18]. In addition, the isolated heart used in the present investigation is a blood free experimental model, thus, no circulating hormones and transmitters are present, which influence the response of cardiac tissue to different stimuli [27]. Finally, it should be also noted that a direct test for electrical instability following VF episodes was not carried out [37].

## Figures and Tables

**Figure 1 ijms-20-01628-f001:**
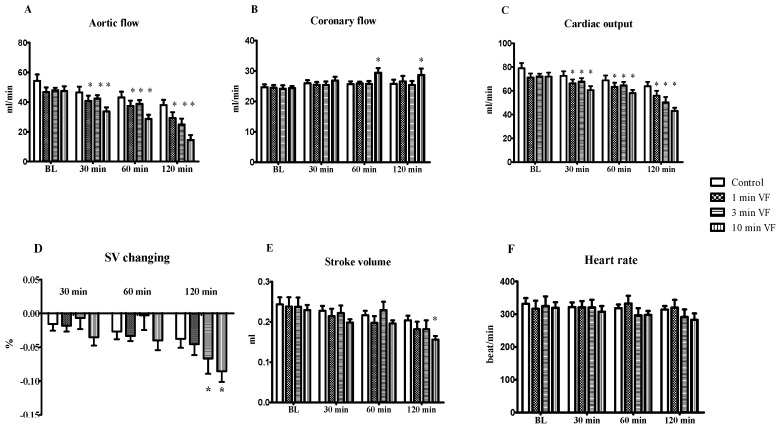
Effects of electrically induced ventricular fibrillation (VF) and post-fibrillation recovery on cardiac function in isolated working heart preparation. Isolated working hearts taken from ketamine/xylazine-anesthetized Sprague-Dawley rats and segregated into test groups of 6 hearts per group were subjected to 0 (Control), 1, 3, and 10 min periods of electrically-induced VF (20 Hz, 1200 beats/min), respectively. Each heart was then perfused with oxygenated Krebs-Henseleit bicarbonate buffer, and then allowed post-fibrillation recovery periods for 30, 60, and 120 min. Cardiac function was registered following each electrical VF induction and recovery period, including aortic flow (**A**); coronary flow (**B**); cardiac output (**C**); stroke volume changing (**D**); stroke volume (**E**); and heart rate (**F**). Results are shown as average values from each group of rat hearts ± SEM. *n* = 6 in each group, * *p* < 0.05 in comparison with the time-matched control values.

**Figure 2 ijms-20-01628-f002:**
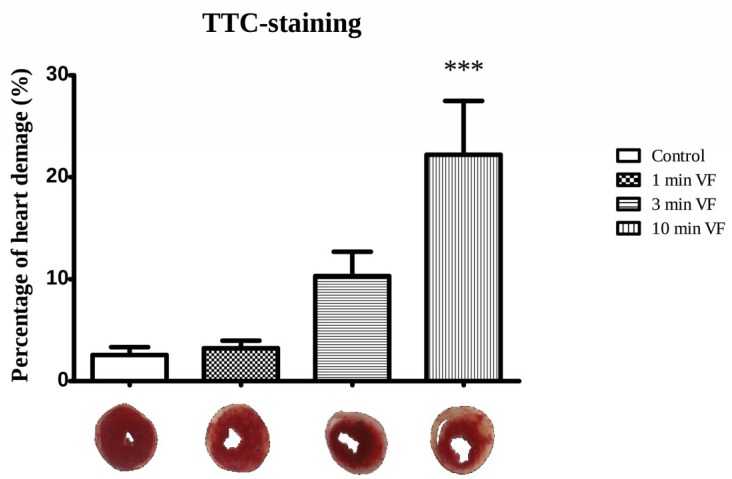
Effects of electrically induced ventricular fibrillation on infarcted size. The time-dependent effects of electrically induced VF duration (1, 3, and 10 min) followed by 120 min of perfusion on infarcted zone extent in isolated working hearts are shown. Hearts taken from ketamine/xylazine-anesthetized Sprague-Dawley rats and segregated into test groups of 6 hearts per group, were subjected to 0, (Control), 1, 3, and 10 min periods of electrically-induced VF (20 Hz, 1200 beats/min), respectively, followed by 120 min of perfusion. Staining, to demonstrate infarcted areas of heart tissue, was accomplished with triphenyl tetrazolium chloride (TTC) solution in phosphate buffer. The percentages of heart damage were assessed in 2–3 mm thick sections transversely sliced from the apico-basal axis into 2–3 mm thick sections and analyzed using ImageJ software. Infarcted areas were measured by computerized planimetry software and infarct magnitude was determined as percentage ratio of the infarcted zone to the total area in each heart (percentage of pixels). Results are shown as average values from each group of rat hearts ± SEM, and in comparison, with the control value. *n* = 6 in each group, *** *p* < 0.001.

**Figure 3 ijms-20-01628-f003:**
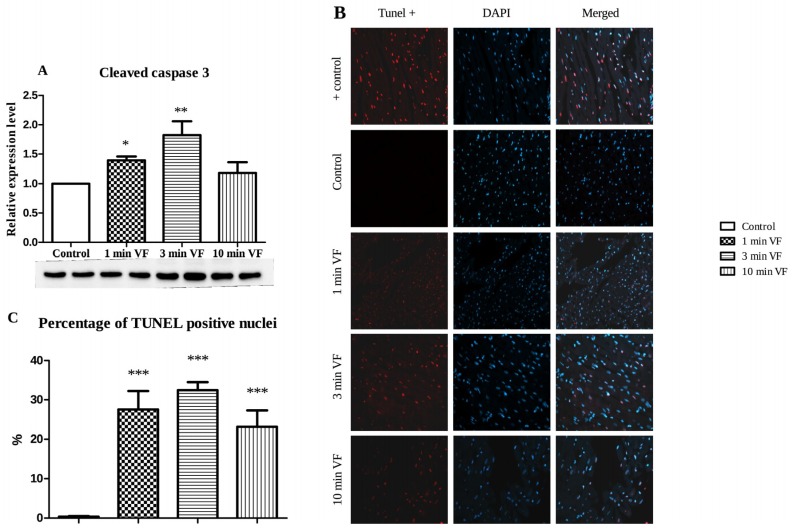
Western blot analysis: Effects of electrically induced fibrillation followed by aerobic perfusion on heart tissue expression of cleaved-caspase-3. Cleaved caspase-3 expression (**A**), representative TUNEL positivity (**B**), and percentage of TUNEL positive nuclei (visualized by a Zeiss AxioScope A1 microscope, EC Plan-Neo fluar 40×/0.75 M27 objective lens with HBO100 illuminator and Zeiss AxioCam ICm1 camera (Zeiss, Jena, Germany) (**C**) are depicted. *n* = 8 in each group, mean ± SEM, comparisons were made to the fibrillation-free control group. * *p* < 0.05, ** *p* < 0.01, and *** *p* < 0.01.

**Figure 4 ijms-20-01628-f004:**
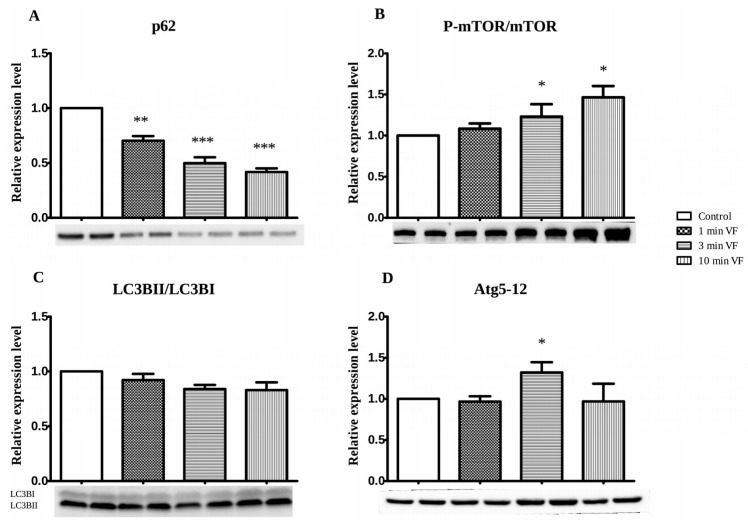
Western blot analysis: Effects of electrically induced fibrillation on heart tissue expression of p62 (**A**), mTOR-phosphorylated, and mTOR (mTOR) (**B**) ratio, LC3BII/LC3BI ratio (**C**) and Atg5-12 complex (**D**) proteins. The time-dependent (0, 1, 3, and 10 min) effects of electrically induced VF followed by 120 min perfusion on expressions of selected proteins in isolated working hearts are shown. Hearts were taken from rats and segregated into test groups of 6 hearts per group, and were subjected to 0 (Control), 1, 3, and 10 min periods of electrically-induced VF (20 Hz, 1200 beats = min), respectively, followed by 120 min of aerobic perfusion. Western blot analysis to assess expression of each target protein was conducted. Signals for each protein band were measured using Bio-Rad Clarity Western ECL Substrate and optical density of the bands was evaluated with the ChemiDoc Touch Imaging System. Protein expressions for each sample of heart tissue was calculated using Bio-Rad Image Lab 5.2 software. Results are shown as average values from each group of 6 hearts, mean ± SEM. * *p* < 0.05, ** *p* < 0.01, and *** *p* < 0.001 compared to the Control (non-fibrillated group).

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
