# Peer review of "Electrically-Induced Ventricular Fibrillation Alters Cardiovascular Function and Expression of Apoptotic and Autophagic Proteins in Rat Hearts"

_ijms, 2019, doi:10.3390/ijms20071628_

Round 1

Reviewer 1 Report

The present article aims to evaluate the effect of electrically stimulated ventricular fibrillation on autophagy and apoptosis in isolated working hearts.

A few minor revisions are needed, as follows:

Please correct spelling of some words, starting with line 37 “demage” and line 55 “orinciple”.

Abstract: Please mention also changes of AF, CF, CO and SV.

Introduction, line 51: You mention “likelihood of recovery” but did not test electrical instability after VF episodes. There are several ECG variables able to assess ventricular arrhythmia risk, such as QT and Tpeak-Tend intervals (Mozos I, Caraba A. Electrocardiographic predictors of cardiovascular mortality. Dis Markers. 2015; 2015:727401.doi: 10.1155/2015/727401).  Please mention it as a study limitation.

Results: Please provide the exact p values for the paragraph 2.1. Effects of pacing induced ventricular fibrillation on cardiac recovery!

2.1, lines 96-99: Please rephrase!

Discussion should start with a phrase emphasizing your findings.

Discussion, lines 162-164: You state: “Depression in cardiac function including aortic flow and cardiac output were directly proportion to the duration of electrical-stimuli induced ventricular fibrillation.” Please provide a proof for your statement and rephrase!

Discussion, lines 181-185: Please check again because it is confusing! Is it about ischemic preconditioning or pacing?

Study limitations: Markers of oxidative stress were not assessed. Please mention the limits of the isolated working heart.

Your Conclusions belong to Discussion. Please replace the part included for the moment in Conclusion with your own findings!

Author Response

The present article aims to evaluate the effect of electrically stimulated ventricular fibrillation on autophagy and apoptosis in isolated working hearts.

A few minor revisions are needed, as follows:

Please correct spelling of some words, starting with line 37 “demage” and line 55 “orinciple”.

The typos have been corrected through the entire manuscript in the revised version.

Abstract: Please mention also changes of AF, CF, CO and SV.

We have included in the Abstract, as the reviewer suggested.

Introduction, line 51: You mention “likelihood of recovery” but did not test electrical instability after VF episodes. There are several ECG variables able to assess ventricular arrhythmia risk, such as QT and Tpeak-Tend intervals (Mozos I, Caraba A. Electrocardiographic predictors of cardiovascular mortality. Dis Markers. 2015; 2015:727401.doi: 10.1155/2015/727401).  Please mention it as a study limitation.

The following section has been inserted:

The limitation of the study was inserted at the end of the Discussion as the followings:

“It has to be noted that direct measurement of oxidative stress markers was not assessed in the present study. However, it has been previously shown that pacing-induced VF induces free radical formation (18). In addition, the isolated heart used in the present investigation is a blood-free experimental model, thus, no circulating hormones and transmitters are present, which influence the response of cardiac tissue to different stimuli (27). Finally, it should be also noted that direct test for electrical instability following VF episodes was not carried out (37).

Results: Please provide the exact p values for the paragraph 2.1. Effects of pacing induced ventricular fibrillation on cardiac recovery!

P values are provided in the revision.

2.1. lines 96-99: Please rephrase!

The section is rephrased as the followings:

As it can be seen in Fig. 2. no significant alteration in infarct volume was observed in hearts subjected to 1 min and 3 min of pacing-induced VF followed by 120 min of aerobic perfusion in comparison with hearts received no electrically induced VF (control).

Discussion should start with a phrase emphasizing your findings.

Discussion, lines 162-164: You state: “Depression in cardiac function including aortic flow and cardiac output were directly proportion to the duration of electrical-stimuli induced ventricular fibrillation.” Please provide a proof for your statement and rephrase!

 This was rewritten in the revised version as the following:

 “The duration of electrical stimuli induced ventricular fibrillation could be contributed to the depression of cardiac function including aortic flow, coronary flow and stroke volume changes.

Discussion, lines 181-185: Please check again because it is confusing! Is it about ischemic preconditioning or pacing?

It has been omitted.

Study limitations: Markers of oxidative stress were not assessed. Please mention the limits of the isolated working heart.

The limitation of the study was inserted at the end of the Discussion as the followings:

“It has to be noted that direct measurement of oxidative stress markers was not assessed in the present study. However, it has been previously shown that pacing-induced VF induces free radical formation (18). In addition, the isolated heart used in the present investigation is a blood-free experimental model, thus, no circulating hormones and transmitters are present, which influence the response of cardiac tissue to different stimuli (27). Finally, it should be also noted that direct test for electrical instability following VF episodes was not carried out (37).

Your Conclusions belong to Discussion. Please replace the part included for the moment in Conclusion with your own findings!

The Conclusion has been inserted to the Discussion as this reviewer suggested.

Reviewer 2 Report

The manuscript titled «Electrically-induced ventricular fibrillation alters cardiovascular function and expression of apoptotic and autophagic proteins in rat hearts» from Andras Czegledi et al. contain new and prior data about modulation of cardiac function by ventricular fibrillations. This is good experimental work with combination of different methods. Overall, the findings of Czegledi et al. demonstrate how ventricular fibrillation influences the level of apoptosis- and autophagy-associated proteins. However, to consider the manuscript suitable for publication in Int. J. Mol. Sci. the authors should address the following specific points.

Page 3. Legend to Fig.1. Misprint in describing the stroke volume (1.D. >> Fig.1E) – and lack of the indication for SV changing (Fig.1D).

Fig.5. Why authors used relative expression level? and did not normalize the bands to GAPDH, for example? Please, clarify this point. And in Methods there is no description of TUNEL experiments.

Page 6. There is no legend to Fig.4.

Author Response

The manuscript titled «Electrically-induced ventricular fibrillation alters cardiovascular function and expression of apoptotic and autophagic proteins in rat hearts» from Andras Czegledi et al. contain new and prior data about modulation of cardiac function by ventricular fibrillations. This is good experimental work with combination of different methods. Overall, the findings of Czegledi et al. demonstrate how ventricular fibrillation influences the level of apoptosis- and autophagy-associated proteins. However, to consider the manuscript suitable for publication in Int. J. Mol. Sci. the authors should address the following specific points.

Page 3. Legend to Fig.1. Misprint in describing the stroke volume (1.D. >> Fig.1E) – and lack of the indication for SV changing (Fig.1D).

The suggestions have been corrected in the revised version of our manuscript.

Fig.5. Why authors used relative expression level? and did not normalize the bands to GAPDH, for example? Please, clarify this point. And in Methods there is no description of TUNEL experiments.

Stain-free method allows protein quantification against total loaded protein content. Thus, the following sentences have been inserted in the Methods:

“The levels of the investigated proteins were normalized against to total protein loaded on the gels, and the protein expression was quantified by the ratio of (band volume)/(total protein volume). Thus, this method eliminated the need of housekeeping proteins [35,36].

Description of TUNEL is now inserted in the revised version of the manuscript. Please, see: 4.7 Fluorescens cell death detection

Page 6. There is no legend to Fig.4.

The Legend for Figure 4 is described now in details.